# Image Encryption and Decryption System through a Hybrid Approach Using the Jigsaw Transform and Langton's Ant Applied to Retinal Fundus Images

Andrés Romero-Arellano [1,†], Ernesto Moya-Albor [1,*,†], Jorge Brieva [1,†], Ivan Cruz-Aceves [2,†], Juan Gabriel Avina-Cervantes [3,†], Martha Alicia Hernandez-Gonzalez [4,†] and Luis Miguel Lopez-Montero [4,†]





1. Facultad de Ingeniería, Universidad Panamericana, Augusto Rodin 498, Ciudad de México 03920, Mexico; 0228652@up.edu.mx (A.R.-A.); jbrieva@up.edu.mx (J.B.)
2. CONACYT-Centro de Investigación en Matemáticas (CIMAT), A.C., Jalisco S/N, Col. Valenciana, Guanajuato 36000, Mexico; ivan.cruz@cimat.mx
3. Department of Electronics Engineering, Engineering Division of Campus Irapuato-Salamanca, University of Guanajuato, Salamanca 36885, Mexico; avina@ugto.mx
4. Unidad Médica de Alta Especialidad (UMAE), Hospital de Especialidades No. 1 Centro Médico Nacional del Bajío, IMSS, León 37320, Mexico; martha.hernandezg@imss.gob.mx (M.A.H.-G.); luis.lopezmo@imss.gob.mx (L.M.L.-M.)
* Correspondence: emoya@up.edu.mx; Tel.: +52-55-5482-1600 (ext. 5210)
† These authors contributed equally to this work.

**Abstract:** In this work, a new medical image encryption/decryption algorithm was proposed. It is based on three main parts: the Jigsaw transform, Langton's ant, and a novel way to add deterministic noise. The Jigsaw transform was used to hide visual information effectively, whereas Langton's ant and the deterministic noise algorithm give a reliable and secure approach. As a case study, the proposal was applied to high-resolution retinal fundus images, where a zero mean square error was obtained between the original and decrypted image. The method performance has been proven through several testing methods, such as statistical analysis (histograms and correlation distributions), entropy computation, keyspace assessment, robustness to differential attack, and key sensitivity analysis, showing in each one a high security level. In addition, the method was compared against other works showing a competitive performance and highlighting with a large keyspace ($>1 \times 10^{1,134,190.38}$). Besides, the method has demonstrated adequate handling of high-resolution images, obtaining entropy values between 7.999988 and 7.999989, an average Number of Pixel Change Rate (NPCR) of 99.5796% ± 0.000674, and a mean Uniform Average Change Intensity (UACI) of 33.4469% ± 0.00229. In addition, when there is a small change in the key, the method does not give additional information to decrypt the image.

**Keywords:** medical image; image encryption and decryption; Jigsaw transform; Langton's ant; deterministic noise; retinal fundus images

## 1. Introduction

Nowadays, digital medical images such as X-ray radiography, ultrasound, magnetic resonance imaging or computed tomography play an important role in diagnosis and treating diseases. They are used in many modern hospitals and they involve patients' private information as well as confidential and sensitive data [1,2].

On the other hand, there are several applications that require storing and transmitting medical images over vulnerable to security attack channels, for example, the internet. Therefore, it is necessary to protect the patients' information including the medical im- ages data. In this sense, cryptography, a sub-field of mathematics and computer science, has generated schemes such as the standards DES (Data Encryption Standard) and AES (Advanced Encryption Standard), which have been used to protect text data through and

robust encryption and decryption approaches. However, these approaches have some disadvantages for large volume data present in medical images, such as a high computing time, high redundancy, and a strong correlation in neighbor pixels [2].

In different medical technologies such as Telemedicine and Tele-surgery, the use, storage, and transmission of ultrasound, computed tomography, retinal fundus image, and other images plays a vital role. Moreover, the network transmission of these images, via the internet or the hospital intranet, between doctors, specialists, or researchers poses serious security problems, making them vulnerable to malicious tampering and privacy leakage. Therefore, the development of efficient medical image encryption methods is an active area.

On the other hand, both in natural and medical images, the encryption methods must try to preserve the quality of the decrypted image concerning the original one. The above becomes more relevant in medical diagnosis, where the integrity of anatomical or functional information may affect medical interpretation.

Many encryption schemes are based on position scrambling (permutation) to secure image data, other methods use the chaos theory, which has unpredictability and ergodicity properties [2]. Nevertheless, some cryptanalysis works are showing that some encryption schemes have the risks of being broken [3–5]. On the other hand, due to the limitations of the used chaotic systems, for example, the pseudorandom number generators, some chaos-based cryptography systems have low security levels [6,7].

Therefore, it is necessary to develop new cryptography systems that ensure image visual data integrity, being robust against recent cryptanalysis techniques, and improving the current chaos-based cryptography systems.

Some methods combine chaos theory and permutation techniques to improve the encryption results. For example, in [8] the authors proposed a chaos-based image encryption method and imitating the Jigsaw technique as a scrambling scheme, its proposal consists of three steps, a pre-processing, encryption, and post-processing stage, in pre-processing and post-processing stages the hyperchaotic Lorenz system is used to generate the control sequences of revolving and shifting image blocks, whilst in the encryption process, the skew tent system is applied to revolving image blocks. In [9] an image encryption system using the Jigsaw transform (JT) and the iterative finite field cosine transform is presented. Hua et al. [1] presented a medical image encryption scheme, it first inserts random data in the input image, then, two stages scrambling and pixel adaptive diffusion (bitwise XOR and modulo arithmetic) are applied. In [10] a hybrid digital cryptosystem was presented, it uses the Jigsaw transform to scramble the watermark. Then, the watermark was inserted in the DCT (Discrete Cosine Transform) domain of an input image previously scrambled by a chaotic scrambling algorithm. Kanso and Ghebleh [2] proposed a selective chaos-based image encryption method for medical image applications, it consists of a shuffling phase by chaotic cat maps and a masking phase, both block-based. On the other hand, Wang and Xu [11] presented Langton's ant (LA), a cellular automaton, to scramble the image, where through and intertwining logistic map defined the steps and next position of the ant. Additionally, the authors used a Piecewise Linear Chaotic Map (PWLCM) as the final step to diffuse the image. Stoyanov and Kordov [12] proposed an image encryption algorithm based on the pseudo-random bit generators: Chebyshev map and rotation equation. Aryal et al. [13] proposed an integrated model of block-permutation-based encryption using block scrambling, block-rotation/inversion, negative–positive transformation, and the color component shuffling. In addition, a histogram shifting method was adopted as reversible data hiding. Jaroli et al. [14] proposed a color image encryption based on four-dimensional differential equations chaotic map and Arnold map. In [15] Gao et al. presented an encryption scheme based on fractional-order hyperchaotic systems and multi-image fusion, where the authors performed an analysis of the circuit and the dynamic of the chaotic system. Wang and Chen [16] proposed a method for image scrambling and diffusion, which combines one-dimensional (Logistic) and two-dimensional chaotic map systems (2D Logistic-adjusted-Sine) to generate chaotic

sequences. Then, an L-shaped method based on the dynamic block is used to scramble the image, followed by a diffusion stage at the bit level. In [17] Wang and Zhang presented a dynamic encryption algorithm both for the scrambling and diffusion stages. The dynamic behavior is reached by changing the pseudo-random number generated by the chaotic system in each round. The chaotic system consists of a compound one-dimensional nested sine map. Enayatifar et al. [18] reported a 3D chaotic function (3-D logistic map) to generate a synchronous permutation-diffusion encryption method. The first dimension of the logistic map joint with a Deoxyribose Nucleic Acid (DNA) sequence are used to permute the pixel. While that the second and third dimensions are associated with the DNA operator to alter the pixel value. In [19] Ibrahim and Alharbi presented an image encryption scheme based on the Henon map by a dynamic substitution box (S-box) confusion and an elliptic curve cryptosystem. In [20] Azam et al. proposed a fast, public-key, and two-phase image encryption scheme based on elliptic curves. First, the plain text is masked by using random numbers. Then the pixels are scrambled by using a dynamic S-box. Laiphrakpam and Khumanthem [21] presented an image encryption scheme based on a chaotic system and elliptic curve over a finite field. It consists of a chaotic diffusion phase, a substitution phase using S-boxes, a diffusion phase based on the Logistic map, and a block permutation operation. Hayat and Azam [22] proposed a two-phase image encryption method by constructing S-boxes and pseudo-random numbers using a total order on an elliptic curve over a prime field. First, the image is diffused by masking it by the proposed pseudo-random number, which is then confused by a proposed dynamic S-box. Wan et al. [23] proposed an algorithm that uses genetic optimization to optimize chaos parameters, which are then applied to the result of permuting the pixels of an image. Liang et al. [24] reported an improvement to Arnold transform (AT), using a double scrambling encryption algorithm based on AT which modifies both the position of the pixels and their gray values. This method can get the desired results faster than the traditional AT while remaining as practical as the original AT. Kaur et al. [25] presented a watermarking technique combined with RSA (Rivest–Shamir–Adleman) and fractal image coding, which enhances the security against attacks such as cutting, random noise attack, and JPEG compression. Kumar Sinha et al. [26] employed Arnold transformation to produce a confused image and S-box transformation and XOR operation are used to provide diffusion. Ballesteros et al. [27] reported a method that uses variable length codes based on Collatz conjecture for transforming the content of the image into non-intelligible audio. The scrambling and diffusion processes are performed simultaneously in a non-linear way. Swathika et al. [28] proposed a technique based on chaos theory known as Confusion and Diffusion. The confusion step uses block scrambling and modified zigzag transformation and the Diffusion step uses 3D logistic map and key generation followed by the additive cipher. In [29] Folifack et al. studied the dynamic properties of a Jerk system as well as DNA coding with the purpose of the implementation of an encryption technique.

In regard to medical image cryptography, for example the fundus photographs encryption, in the work of Mehta et al. [30], the authors proposed a lossless cryptosystem for fundus images based upon chaotic theory using a combination of scrambling and substitution architecture. On the other hand, Gamal et al. [31] presented a hybrid encryption scheme using chaotic maps and 2D Discrete Wavelet Transform (DWT) steganography, with application to transmit securely retinal fundus medical images. In [32] Moafimadani et al. presented an algorithm based on chaotic systems to protect medical images against attacks. It uses a high-speed permutation process followed by an adaptive diffusion. Javan et al. [33] proposed a medical image encryption method based on multi-mode synchronization of hyper-chaotic systems, where their main contribution was to encrypt medical images based on robust adaptive control. In [34] Xue et al. proposed an image protection algorithm based on the deoxyribonucleic acid chain of dynamic length. The method encrypts the image by DNA dynamic coding, DNA dynamic chain, and dynamic operation of row chain and column chain. The authors tested their method against three kinds of medical images. Kumar and Gupta [35] presented an encryption algorithm for medical images

based on the 1D logistic map associated with pseudo-random numbers. For the logistic map, the authors analyzed the effect of its initial values and parameters, which are involved in the shuffling and substituting processes. In addition, the method was tested both natural as medical images, such as the human brain, MRI, and lungs. In [36] Nematzadeh et al. reported a medical image encryption method based on a modified genetic algorithm and coupled lattice map. The coupled lattice map improves the cipher, while the modified genetic algorithm includes a new local search method and a stop condition, accelerating the convergence. Cao et al. [37] presented a medical image encryption algorithm based on edge maps, which includes a bit-plane decomposition, a generator of a chaotic sequence using a Sine map, and a scrambling method. Sarosh et al. [38] presented an algorithm that circularly shifts the pixels of an image, then the Most Significant Bit (MSB) plane is replaced by a plane resulting from the XOR operation between the MSB plane and the seventh intermediate significant bit plane. The result is then scrambled using pseudo random numbers generated with logistic map. Then, the result is XORed with a key image was generated using the Piecewise Linear Chaotic Map. Finally a Chebyshev map is used to permute the pixels. Salama et al. [39] fused the wavelet-induced multi-resolution decomposition capacity of the Discrete Wavelet Transform with the energy compaction of the Discrete Cosine Transform for a technique that outperforming existing methods in terms of imperceptibility, security, and robustness. Ravichandran et al. [40] proposed an algorithm based in the Integer Wavelet Transform, DNA computing, and shuffling. Ge [41] proposed an encryption algorithm called ALCencryption, which applies an improved Arnold map to gray images using the optimal number of iteration, then the algorithm uses Logistic and Chebyshev map cross-diffusion. This improved Arnold map is generalized for images of any size. Color images are encrypted by cross-diffusion of double chaotic map. Carey et al. [42] presented an algorithm utilizing two biometrics of the user, the iris and the fingerprint, which are hashed through the Indexing First One hashing which are then used as two different keys in a two-round Advanced Encryption Standard Cipher Block Chaining system to encrypt medical images, improving in many existing schemes based on biometrics. Additionally, the process is lossless, which is necessary for a medical encryption system. Li et al. [43] proposed an algorithm for protecting key regions on the image. Firstly, coefficients to measure the variation are used to identify the key regions (when a lesion is present for example), and the texture complexity is analyzed. Then, the data-hiding algorithm embeds lesion area contents into a high-texture area and an Arnold transformation is used to protect the original lesion information. Finally, the authors use image basic information cipher-text and decryption parameters to generate a QR code associated with the original key regions. In [44] Sangavi and Thangavel presented a Multi-dimensional Medical Image Encryption scheme exploiting the chaotic property of the Rossler dynamical system and Sine map. Siddartha et al. [45] proposed an efficient data masking technique based on chaos and on the DNA code used for the encryption for securing the healthcare data images. Chai et al. [46] reported a medical image encryption scheme combining Latin square and chaotic system. Banik et al. [47] proposed an encryption scheme for multiple medical images using an elliptic curve analog ElGamal cryptosystem and Mersenne Twister pseudo-random number generator. This strategy decreases the encryption time and solves the problem of data expansion associated with the ElGamal cryptosystem. In [48] Pankaj and Dua proposed a one-dimensional Tangent over Cosine Cosine (ToCC) chaos map method to encrypt medical images. First, padding is performed on the input image to hide the original dimension. The second step consists of the generation of two different chaotic sequences using the ToCC and Chebyshev-Chebyshev chaotic maps, respectively. In the third stage, a modified High-Efficiency Scrambling (mHES) that uses the ToCC chaotic map-generated sequence is applied to perform first-level scrambling. Finally, modified Simultaneous Permutation and Diffusion Operation (mSPDO) that exploits the Chebyshev-Chebyshev chaotic map is used to implement second-level scrambling to obtain the final encrypted image. Elamir et al. [49] reported an encryption scheme for hiding patient information in medical images.The strategy uses the Least Significant Bit algorithm, involving

hiding data in the least bit of image pixels. The image is then compressed with a key generated by chaotic maps and DNA encoding rules.

In this paper, we propose a new hybrid encryption system applied to high-resolution fundus photographs. It uses Jigsaw transforms and cyclic permutations to scramble the image hiding the visual information. Additionally, it uses Langton's ant and a novel deterministic noise algorithm to obtain a high-level secure encrypting image. To test the performance of the proposed method, we performed several tests over the encrypted image, including statistical analyses as the histogram comparison and pixel neighborhood correlation, entropy computing, the keyspace universe determination, a differential attack testing, and a key sensitivity studying.

The rest of the paper is organized as follows: Section 2.1 presents the medical image dataset used in this work. Section 2.2 describes the Jigsaw transform, Section 2.3 presents an image spatial cyclic permutation technique, the Langton's ant concept is shown in Section 2.4, and Section 2.5 defines a deterministic noise algorithm. Sections 2.6 and 2.7 develop the image encryption and decryption proposal using the Jigsaw transform, the deterministic noise and Langton's ant. The experimental results are presented in Section 3. Section 4 presents a discussion about the results obtained in this work, as well as a comparison with other works. Finally, Section 5 concludes the paper and presents future work.

## 2. Materials and Methods

### 2.1. Dataset Description

The image dataset used in this paper is composed of 20 RGB fundus photographs, 10 from healthy patients and 10 from ill patients with retinopathy. Figure 1 shows the complete image dataset, where each image has a spatial resolution of $4000 \times 6000$ pixels with 24 bits per pixel. The first two rows in Figure 1 correspond to healthy patients and the last two rows to non-healthy patients. In each original image, firstly, we scanning the rows and columns to delimited an ROI (Region of Interest) containing the area closer to the eyeball, removing a large area of black pixels around it. At present, the dataset is not publicly available.

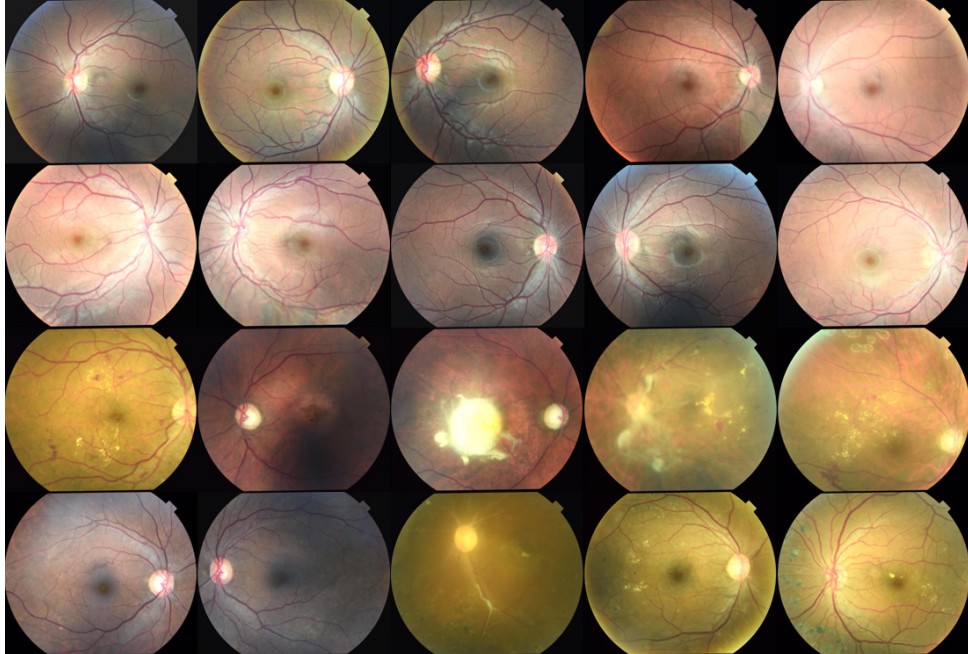

**Figure 1.** Complete image dataset. Healthy patients (rows 1 and 2). Non-healthy patients (rows 3 and 4).

### 2.2. Jigsaw Transform

The Jigsaw transform ($J\{\}$) is a nonlinear operator which randomly juxtaposes different sections of a complex image. It has the advantage of encrypting and decrypting the image using the same algorithm [50].

Let $I(x, y, \tau)$ a multi-spectral digital image $N$ bands, with $x, y$ as its spatial coordinates and $\tau = \tau_1, \ldots, \tau_N$ representing the index for the $N$ bands.

First, the $I(x, y, \tau)$ image is broken up into $M$ non-overlapping subsections of $k1 \times k2$ pixels for $x$ and $y$ in all bands. Then, each block is relocated using some random permutation. The Jigsaw transform is unitary, which holds the energy both for the direct $J_M\{\}$ and the inverse transform $J_{-M}\{\}$. Thus, it satisfies the relation shown in Equation (1):

$$I(x, y, \tau) = J_{-M}\left\{ J_M\left\{ I(x, y, \tau) \right\} \right\}. \tag{1}$$

Figure 2 shows a RGB image (Lena image) and the Jigsaw transform result for $M = 16, 64, 1024$, generating resulting non-overlapping subsections of $64 \times 64$, $32 \times 32$, and $8 \times 8$ pixels respectively.

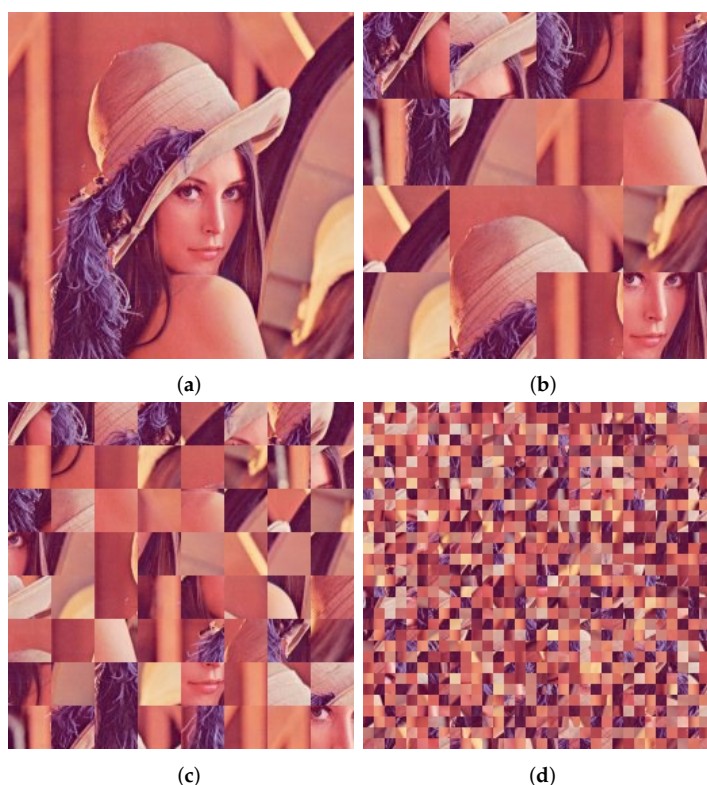

**(a)**     **(b)**

**(c)**     **(d)**

**Figure 2.** Examples of the Jigsaw transform on an RGB image by varying the number of subsections ($M$). (**a**) RGB image. (**b**) $M = 16$, blocks of $64 \times 64$ pixels. (**c**) $M = 64$, blocks of $32 \times 32$ pixels. (**d**) $M = 1024$, blocks of $8 \times 8$ pixels.

### 2.3. Image Spatial Cyclic Permutation

A cyclic permutation (CP) shifts all elements of a finite set $S$ composed of $L$ elements by a offset $k$, where the elements from the end (or beginning) of the set are inserted at the beginning (or ending) in a cycle way, this mapping operation can be written as $a_i \rightarrow a_{i+k \pmod L}$, where $k \pmod L$ is the modulo operation [51]. Let $A = \{a_0, a_1, \ldots, a_{L-1}\}$, where for $k \geq 1$ we would obtain a right cyclic permutation and for $k \leq 1$ a left cyclic permutation. Thus, a cyclic permutation of one place to the right ($k = 1$) would generate the set $A_{k=1} = \{a_{L-1}a_0, a_1, \ldots\}$ and similarly for $k = -1$ (left cyclic permutation) we would obtain the set $A_{k=-1} = \{a_1, \ldots, a_{L-1}, a_0\}$.

For the case of a multi-spectral digital image $I(x, y, \tau)$, we can applied the cyclic permutation $CP\{k_x, k_y\}$ both to the spatial coordinate $x$ and $y$ by the mapping operation of Equation (2):

$$CP\{k_x, k_y\} = I\left(x + k_x \ (\text{mod } X), y + k_y \ (\text{mod } Y), \tau\right), \tag{2}$$

where $X$ and $Y$ are the horizontal and vertical dimensions of $I(x, y, \tau)$; and $k_x$ and $k_y$ are the horizontal and vertical offsets, respectively.

In Figure 3a, we show the Lena image, a multi-spectral digital image $I(x, y, \tau)$ of $256 \times 256$ pixels, Figure 3b shows the left horizontal cyclic permutation $CP\{k_x = -150, k_y = 0\}$ applied, Figure 3c the vertical upward cyclic permutation $CP\{k_x = 0, k_y = 150\}$ resulting, and Figure 3d the left cyclic permutation $CP\{k_x = -100, k_y = 0\}$ followed by a upward cyclic permutation $CP\{k_x = 0, k_y = -100\}$ over the image.

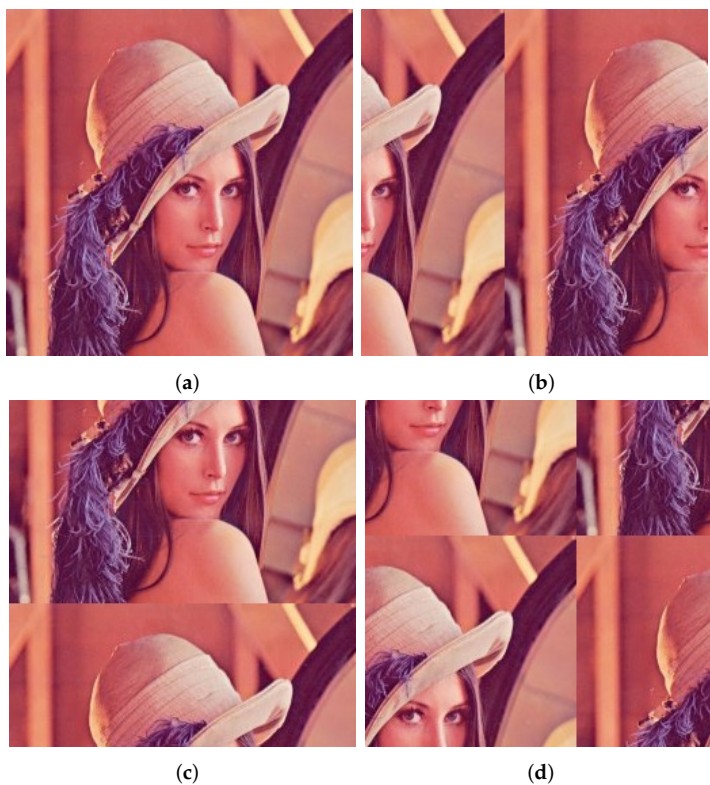

**Figure 3.** Examples of a cyclic permutation over the Lena image. (**a**) RGB image of Lena. (**b**) Left horizontal cyclic permutation $CP\{k_x = -150, k_y = 0\}$ result. (**c**) Vertical upward cyclic permutation $CP\{k_x = 0, k_y = 150\}$ result. (**d**) Left cyclic permutation $CP\{k_x = -100, k_y = 0\}$ followed by a upward cyclic permutation $CP\{k_x = 0, k_y = -100\}$ applied.

### 2.4. Langton's Ant

Langton's ant is a cellular automaton invented in the 1980s by Christopher Langton [52]. It has been studied in many fields as artificial life, computational complexity, cryptography, emergent dynamics, Lorents lattice gas, etc. In particular, its study has been motivated due to the hardness of predicting its macroscopic behavior starting from a determinate microscopic initial configuration [53].

Langton's ant is a 2D universal Turing machine with two main characteristics: (i) a simple set of rules and (ii) a complex emergent behavior. Langton's ant consists of an infinite grid of cells, which have two states: ON and OFF. An imaginary ant is placed in one of the cells, which will walk through the grid following two rules: (i) if it is in a cell turned OFF, it will rotate 90 degrees clockwise, turn ON the cell and move to the cell in front of it. (ii) On the other hand, if it is in a cell that is turned ON, it will rotate 90 degrees

counterclockwise, turn OFF the cell, and move to the cell in front of it. The first four steps of an ant following these rules is shown in Figure 4.

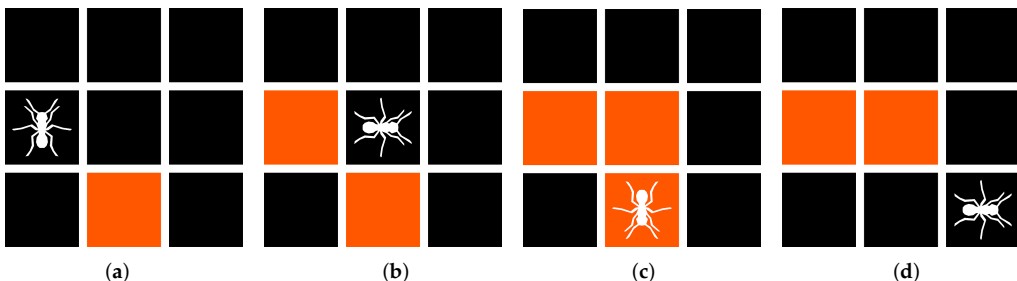

(**a**)       (**b**)       (**c**)       (**d**)

**Figure 4.** The first 4 iterations of an ant on a 3 × 3 grid. The black cells are OFF, the rest are ON. (**a**) First iteration. (**b**) Second iteration. (**c**) Third iteration. (**d**) Fourth iteration.

Initially, the ant's behavior seems chaotic, but eventually the ant stabilizes in a 104-step pattern called "the highway" that will continue indefinitely unless there is a turned ON cell in its path as shown in Figure 5. This behavior appears when the ant is placed on the all-white vertices, so the ant stays in an area of 45 × 48 by around 10,000 steps, and unpredictably it starts dancing a highway of 104 steps. In the highway pattern, the ant moves diagonally with a speed of 2/52. For years this phenomenon has taken place with no exception, generating the highway conjecture: "By starting from any initial configuration with finite support, the highway must eventually appear" [53].

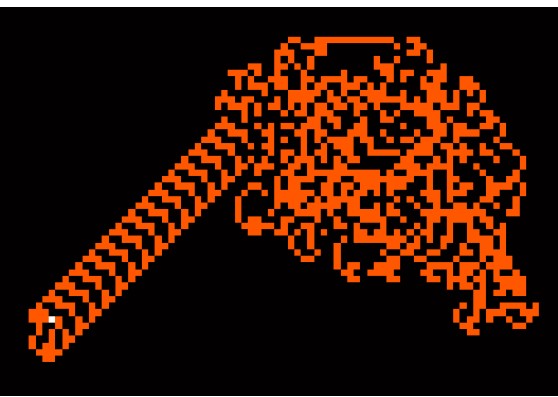

**Figure 5.** The ant (white cell), after 11,538 iterations, is stuck in "The Highway".

The above rules apply only to an infinite grid, if it is finite the rules must be modified to tell the ant what to do when it encounters an edge, as is the case in digital images.

In the work of Wang and Xu [11], the authors used the originals rules of Langton's ant generating the values of the grid through an intertwining logistic map and performing an adjustment of coordinates and rotation directions due to the finite grid of a digital image.

In our case, we will consider that if the ant has to cross one edge it will reappear on the opposite edge, topologically equivalent to the ant being on a torus surface. In this way, if the highway is generated, the pattern will be interrupted because its path will always have obstacles, causing the ant to continue its chaotic movement.

It is trivial that the ant's movement is reversible. To reverse its trajectory it is only necessary to locate the final position and orientation of the ant, rotate it 180 degrees, and let it walk the same number of steps that it originally took.

Unlike Wang and Xu [11], where the original rules were used for digital images, we considered the gray level of the image, thus, for an image of 8 bits (with values between 0 and 255), first we will separate the color channels of the image, applying the ant to each channel, and then we will apply the rules. We have adapted the rules since each channel has 256 possible values in each cell instead of 2 states as in the original version of Langton's

ant: (i) If the ant is at an even pixel it will move as if it were a turned OFF cell, (ii) if the pixel is odd it will move as if it were a turned ON cell. To change the state of the current pixel 47 will be added to the value of the pixel, thus changing it's parity. If the result of the sum is greater than 256, the modulo 256 of the result is taken. When the ant has already walked through the three channels, we put them together to form the encrypted image. To decrypt the image, we separate the channels, let the ant walk from the final coordinates it had during the encryption on each channel, rotated 180 degrees, and taking the same number of steps, and put the channels together.

In Figure 6, we show the result of encrypting the Lena image with different amounts of steps. These examples show the ant's main problem: it requires a large number of steps to reach all the pixels in the image. The larger the image, the more steps it will need.

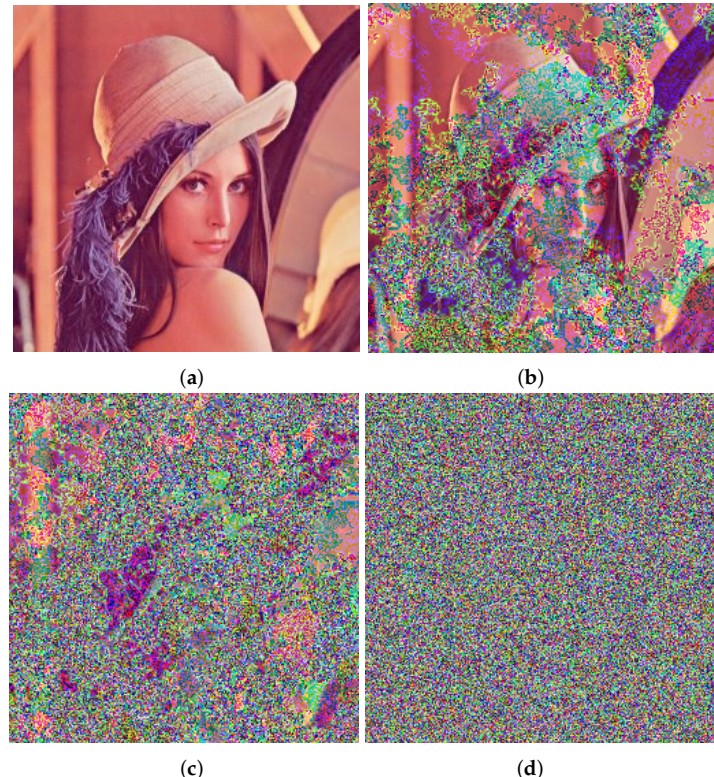

**Figure 6.** Examples of Lanton's ant. (**a**) Input RGB image. (**b**) Result after 100,000 steps. (**c**) Result after 300,000 steps. (**d**) Result after 1,500,000 steps.

It is possible for the ant to enter a highway-like pattern under certain conditions in the initial image, but in any other case the ant will be able to chaotically move an almost arbitrary number of steps.

In this proposal, we will use the ant by dividing the image into four sections (upper left, upper right, lower left, and lower right), repeat these divisions on the resulting sections a $p$ amount of times giving us $4^p$ sections. If the number of pixels of the image is not suitable for dividing it perfectly into those sections, the dimensions of the sections are rounded. Then, we give a starting coordinate for the ant (looking up) and we let it walk some amount of steps on all the sections and on all of their channels. For simplicity we will consider that the ant always takes 100 steps. Even though the starting coordinate will be the same for all sections, given the chaotic movement of the ant they will have different final coordinates and orientations. We save those coordinates and orientations as our decryption key.

### 2.5. Deterministic Noise

Both the Jigsaw transform and spatial cyclic permutation do not modify the image's histogram since they only modify the position of the pixels. On the other hand, Langton's

ant does not completely modify all the pixels because it would have to take a large number of steps. Therefore, to successfully hide the histogram, a possible solution is to add deterministic noise to the image. To achieve this, we design a function that, from three natural numbers, generates pseudo-random natural numbers that will be added to the image.

The details of how the deterministic noise works are the following: given an RGB image $A$ and the RGB image $B$ that will be its version with noise, and given the parameters $p_1, p_2, p_3$, one for each color band, we can calculate the noise that will be added to row $i$. We multiply the three parameters by $i$ and get the variables $z_1, z_2, z_3$ respectively. Next, given the $j$th element of the row of the image $A$ we can calculate $B(i, j, 1)$, $B(i, j, 2)$, and $B(i, j, 3)$ as is shown in Equations (3)–(5):

$$B(i, j, 1) = \mod\left( A(i, j, 1) + \left\lfloor \frac{z_1 * i + j}{z_2 + z_3} \right\rfloor, 256 \right), \tag{3}$$

$$B(i, j, 2) = \mod\left( A(i, j, 2) + \left\lfloor \frac{z_2 * i + j}{z_1 + z_3} \right\rfloor, 256 \right) \text{ and} \tag{4}$$

$$B(i, j, 3) = \mod\left( A(i, j, 3) + \left\lfloor \frac{z_3 * i + j}{z_2 + z_1} \right\rfloor, 256 \right). \tag{5}$$

Since each color channel is an 8-bit image, the resulting value must be between 0 and 255, hence why modulo 256 is used. Next, we will modify the values of $z_1, z_2$, and $z_3$ by calculating first some auxiliary variables $q_1, q_2, q_3$ that will be defined as is given in Equations (6)–(8):

$$q_1 = \mod\left( \left\lfloor \frac{z_1 * i + j}{z_2 + z_3 + 1} \right\rfloor, 256 \right), \tag{6}$$

$$q_2 = \mod\left( \left\lfloor \frac{z_2 * i + j}{z_1 + z_3 + 1} \right\rfloor, 256 \right) \text{ and} \tag{7}$$

$$q_3 = \mod\left( \left\lfloor \frac{z_3 * i + j}{z_2 + z_1 + 1} \right\rfloor, 256 \right). \tag{8}$$

Then we can recalculate $z_1, z_2$, and $z_3$ as observed in Equation (9):

$$z_\tau = \mod(q_\tau + z'_\tau, 256) + 1, \tag{9}$$

where $\tau = 1, 2, 3$ and $z'_\tau$ is the previous value of $z_\tau$.

While we are in row $i$ we continue this process until we have to change to the next row, when this happens we use $p_1, p_2$, and $p_3$ again to calculate $z_1, z_2$, and $z_3$, and continue with the process. In Figure 7 shows the result of applying this deterministic noise to a black RGB picture. It can be seen that, while the picture is clearly affected, there are some areas where the noise does not distort the image completely. The noise seems to work better after the first 300 rows and before 120 columns. Therefore we modified the noise to add 300 to $i$ if its value is less than 300 and to recalculate $z_1, z_2$, and $z_3$ using the parameters $p_1, p_2$, and $p_3$ multiplied by $i$. The result of this modified noise applied to a black image can be seen in Figure 8.

It should be noted that when we are on a determined row, the first time we recalculate the values of $z_1, z_2$, and $z_3$ we get three numbers in the interval $[1, 256]$, and these values determine every next value for the row, which means that after the first pixel of noise has been calculated for the row, there are $256^3$ possible rows that could follow. It should also be noticed that if a set of parameters $p_1, p_2$, and $p_3$ generates a set of $z_1, z_2$, and $z_3$ for a row, and another set of parameters $p'_1, p'_2$, and $p'_3$ generates a the same set of $z_1, z_2$, and $k_3$ for the same row, the variables for the next row could still be different sets. In other words, if we use some parameters and calculate the values for a row, the values for the next rows are not determined by those values. One example is shown in Figure 9 where our deterministic noise is applied two times to a black picture using a different set of parameters $p_1, p_2$, and

$p_3$ in a way that the first row of both results is identical (except obviously for the first element of the row) and the other rows are different.

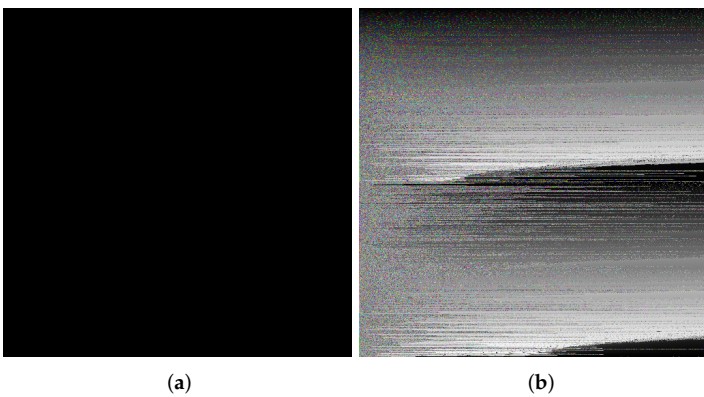

**Figure 7.** The result of the first version of the deterministic noise. (**a**) Black image. (**b**) Deterministic noise added to (**a**).

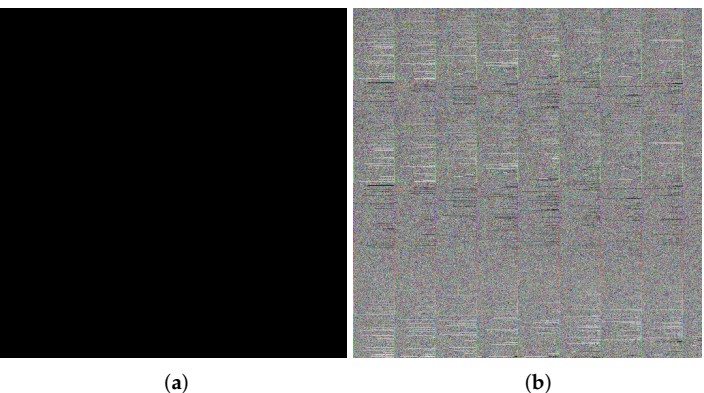

**Figure 8.** The result of the second version of the deterministic noise. (**a**) Black image. (**b**) Deterministic noise added to (**a**).

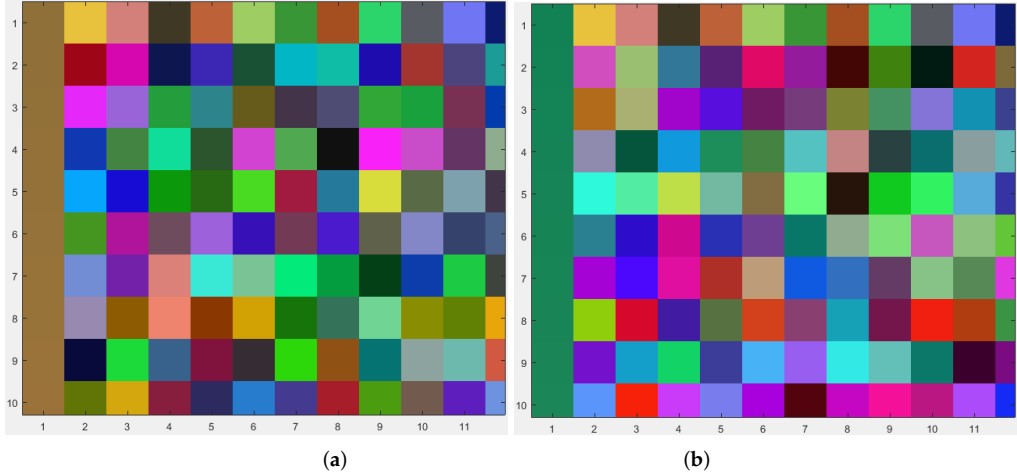

**Figure 9.** Zoom on the first 10 rows and 11 columns of a black picture with deterministic noise added. (**a**) Using the parameters $p_1 = 65$, $p_2 = 31$, and $p_3 = 18$. (**b**) Using the parameters $p_1 = 210$, $p_2 = 133$, and $p_3 = 97$.

To remove this deterministic noise to image $B$ added with the parameters $p_1$, $p_2$, and $p_3$ we do the exact same algorithm but replace Equations (3)–(5) with Equations (10)–(12) respectively:

$$A(i,j,1) = \text{mod}\left(B(i,j,1) - \left\lfloor \frac{z_1 * i + j}{z_2 + z_3} \right\rfloor, 256\right). \tag{10}$$

$$A(i,j,2) = \text{mod}\left(B(i,j,2) - \left\lfloor \frac{z_2 * i + j}{z_1 + z_3} \right\rfloor, 256\right). \tag{11}$$

$$A(i,j,3) = \text{mod}\left(B(i,j,3) - \left\lfloor \frac{z_3 * i + j}{z_2 + z_1} \right\rfloor, 256\right). \tag{12}$$

### 2.6. Encryption Algorithm

Our encryption algorithm uses the previously defined algorithms of the Jigsaw transform, cyclic permutation, Langton's ant and deterministic noise. It can be divided into six steps, as illustrated in Figure 10. The first step is to use the Jigsaw transform (Section 2.2) to scramble the image and hide its visual information. The second step is to add the deterministic noise defined in Section 2.5, hiding most of the histogram of the image. The parameters used to add this noise will be determined by the picture as it will be detailed later. This noise leaves some sections of the image almost unaltered, which could give slight hints of the original colors of the picture. That is why it is needed to re-scramble the image to add more noise. The third step then is to use cyclic permutation, described in Section 2.3, on the image; and the fourth is to use another Jigsaw transform as a consequence of using the cyclic permutation we scramble a different set of pixel blocks in this second Jigsaw transform. The fifth step is to add more deterministic noise (using now parameters given by the user). Finally, the sixth step is to use Langton's ant (Section 2.4).

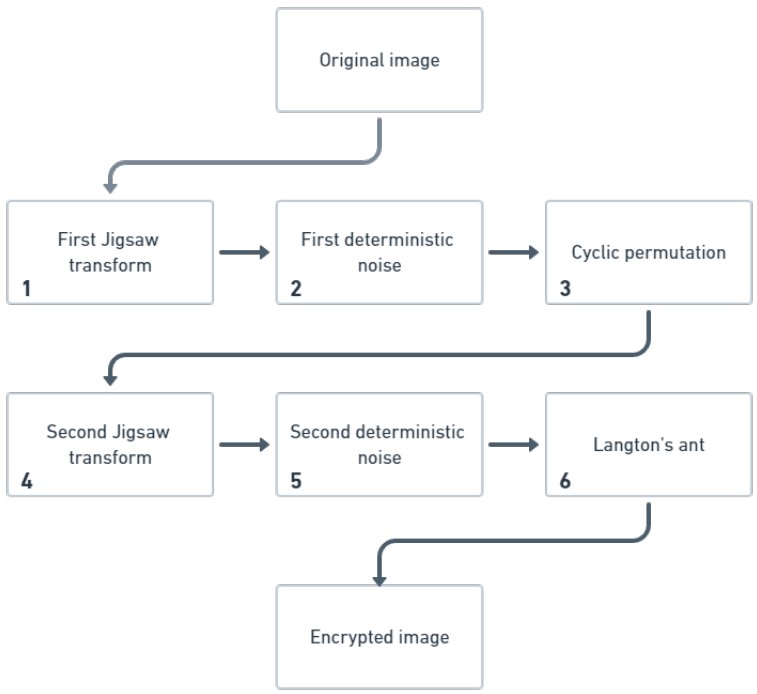

**Figure 10.** The six steps of the encryption algorithm.

The parameters $p_1$, $p_2$, and $p_3$ used for the first deterministic noise on an $X \times Y$ RGB image $I(x,y,\tau)$ are defined as is shown in Equation (13):

$$p_\tau = \text{mod}\left(\sum_{i=1}^{X}\sum_{j=1}^{Y} A(i,j,\tau)(jX - X + i), XY\right), \tag{13}$$

where $\tau = 1, 2, 3$.

This algorithm guarantees that encrypting two images different only in the value of one pixel will have very different results, since they will have a different deterministic noise applied to them.

The parameters needed to decrypt the image are the block size used for JT, the original index of each block of the first JT, the original index of each block of the second JT, the three parameters used for the first deterministic noise, the three parameters used for the second deterministic noise, the two parameters used for the cyclic permutation, and a key for each one of the $4^p$ ants used, each key containing the final coordinate of the ant and it's previous coordinate (to indicate orientation), for each of the three color channels.

### 2.7. Decryption Algorithm

The decryption algorithm uses the inverse function of all the algorithms used for encrypting as illustrated in Figure 11. First, we use the final coordinates and orientations of all the ants from step 6 and we apply Langton's ant to all the sections of the image using those parameters with the ant rotated 180 degrees before. Next, we use the inverse of the deterministic noise using the same parameters that were used for adding the noise. We then perform the inverse of the Jigsaw transform of step 4, the inverse of the cyclic permutation of step 3, the inverse of the deterministic noise of step 2, and finally the inverse of the Jigsaw transform of step 1.

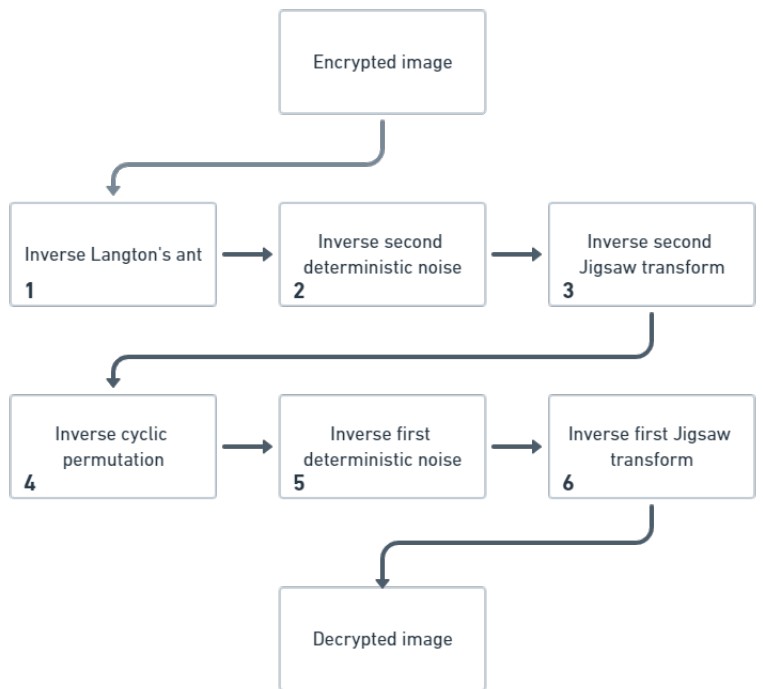

**Figure 11.** The six steps of the decryption algorithm.

### 3. Results

This section presents the results of the proposed hybrid encryption system on high-resolution fundus photographs. We divided it into six stages: Section 3.1 shows some results of the encryption/decryption system for both healthy and non-healthy patients, whilst Section 3.2 presents a statistical analysis between the encrypted and original image, including visual comparison of histograms and the correlation calculation of neighboring pixels, Section 3.3 shows an entropy analysis of the encrypted image, Section 3.4 defines the keyspace universe of the proposed system, Section 3.5 presents a differential attack testing, and finally, Section 3.6 shows a key sensitivity studying.

The encryption and decryption results were obtained on a PC AMD Ryzen 5 3500U running at 21,000 MHz with 12 GB of RAM. The algorithm of encryption has a time-

consuming of 152.58 s and 167.24 s for the decryption algorithm using a fundus photograph of $4224 \times 3616$ over eight cores using parallel computing. For smaller images, the calculation time is considerably reduced, thus using the same equipment and a $512 \times 512$ image, subdividing it into $4^6$ sections for Langton's ant, it takes 1.8694 s to encrypt and 1.8496 to decrypt. When a $256 \times 256$ image is subdivided into $4^5$ sections, it takes 0.5153 s to encrypt and 0.5171 to decrypt. The number of subsections for Langton's ant was chosen to get subsections of a similar dimension to the size of the subsections used for the fundus pictures (around 64 pixels).

### 3.1. Encryption Results

In this section, we show some results of the proposed encryption scheme. For this, we encrypted 2 images, image number 6 from the healthy patients and image number 15 from the sick subset. We used subsections of $16 \times 16$ for the Jigsaw transform, a cyclic permutation of 2005 columns and 2007 rows, $4^9$ sections for Langton's ant (placing the ants of the red channels on the first row and second column, the ants of the green channels on the second row and second column, and the ants of the blue channels on the third row and second column) and $p_1 = 530, 530, p_2 = 120, 120, p_3 = 350, 350$ as the parameters for the second deterministic noise. The results obtained are shown in Figures 12 and 13, respectively.

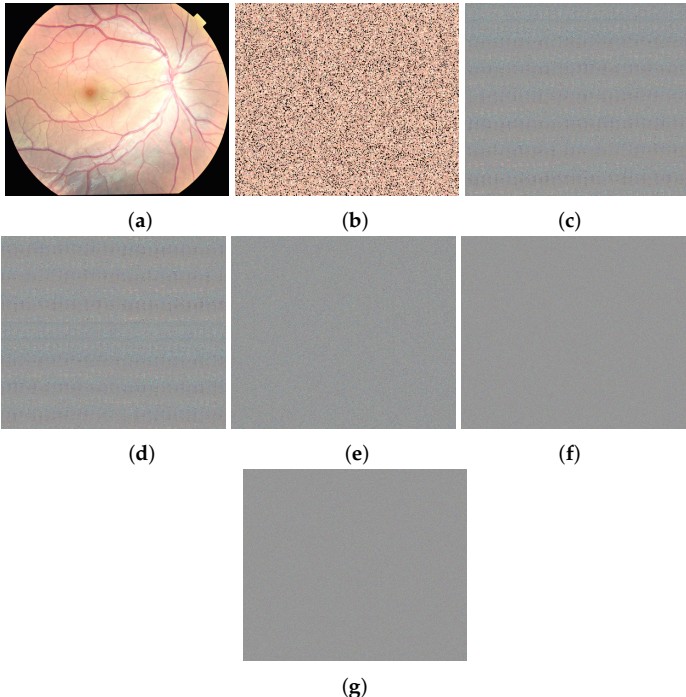

**Figure 12.** Results of the encryption algorithm applied to image 6 from the healthy patients. (**a**) Original image. (**b**) First step: Jigsaw transform. (**c**) Second step: deterministic noise. (**d**) Third step: cyclic permutation. (**e**) Fourth step: Jigsaw transform. (**f**) Fifth step: deterministic noise. (**g**) Sixth step: Langton's ant.

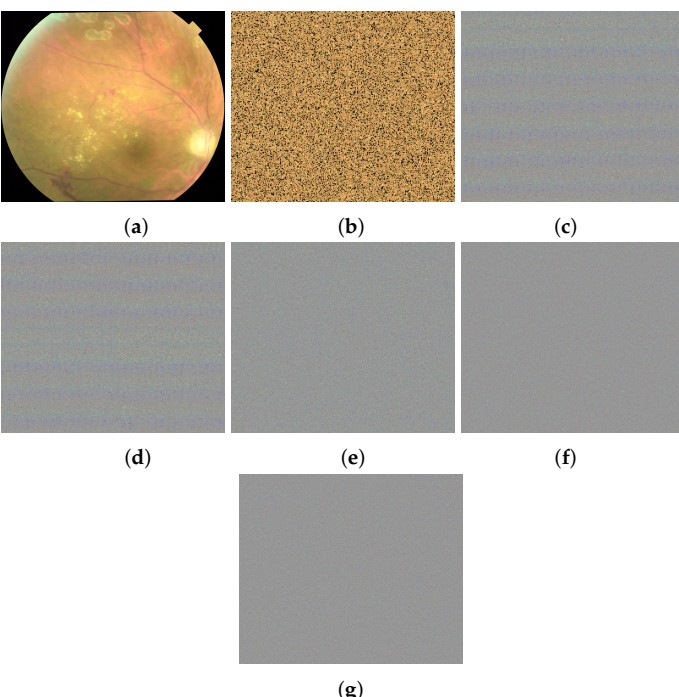

**Figure 13.** Results of the encryption algorithm applied to image 15 from the sick patients. (**a**) Original image. (**b**) First step: Jigsaw transform. (**c**) Second step: deterministic noise. (**d**) Third step: cyclic permutation. (**e**) Fourth step: Jigsaw transform. (**f**) Fifth step: deterministic noise. (**g**) Sixth step: Langton's ant.

It is necessary to say that the Root Mean Square Error (RMSE) between the decrypted and the original images is zero in all cases, showing that the encryption/decryption process is fully reversible when the security key is known.

### 3.2. Statistical Analysis

This section presents a statistical analysis of the results of the proposed encryption method. First, we show the histograms of each channel before and after the encryption process, both of original and encrypted image, allowing us visually comparing the global decorrelation between the intensity levels. Second, we analyze the correlation of neighboring pixels to evaluate the grade of local dependence of pixels in the encrypted image.

### 3.2.1. Histogram Comparison

Due to the nature of the fundus images, both for healthy and non-healthy patients, where the tone and saturation change drastically in each one of them (see Figure 1), It is necessary to carry out a visual comparison between the histograms before and after the encryption process. For this purpose, we used image number 6 from healthy patients and image number 15 from sick patients, which have different histograms, allowing us to qualitatively evaluate the flatness of the resulting histograms concerning the original ones. In this way, in Figures 14 and 15 we show the set of histograms before and after the encryption process for the healthy patient and non-healthy patient, respectively.

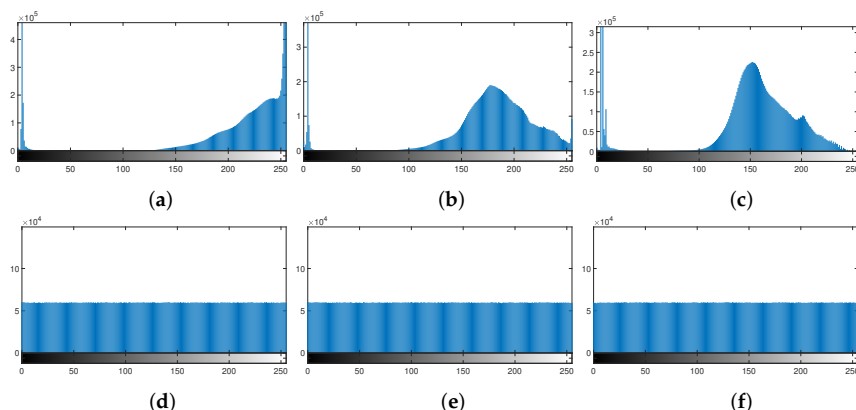

**Figure 14.** Histograms of image number 6 (healthy patient). (**a**) Original image, red channel. (**b**) Original image, green channel. (**c**) Original image, blue channel. (**d**) Encrypted image, red channel. (**e**) Encrypted image, green channel. (**f**) Encrypted image, blue channel.

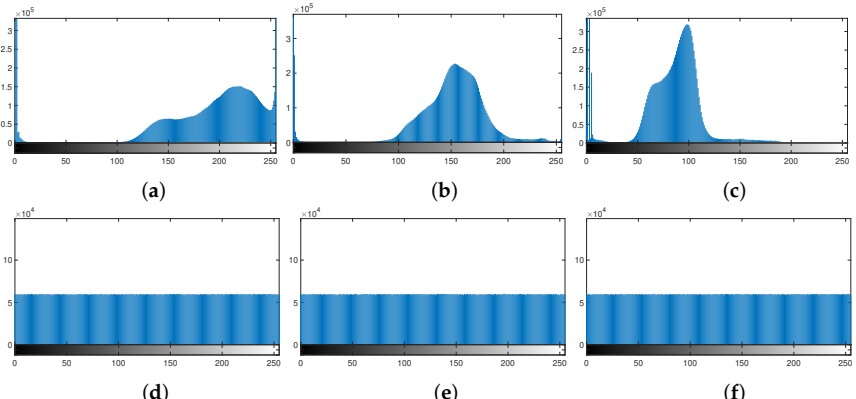

**Figure 15.** Histograms of image number 15 (sick patient). (**a**) Original image, red channel. (**b**) Original image, green channel. (**c**) Original image, blue channel. (**d**) Encrypted image, red channel. (**e**) Encrypted image, green channel. (**f**) Encrypted image, blue channel.

### 3.2.2. Correlation Distributions

The histogram flatness analysis presented in Section 3.2.1 only shows the global decorrelation level of the intensify levels in each channel in the encrypted image, however, it is also necessary to assess the degree of intensity local dependence of neighboring pixels. The above can be measured by computing the correlation of adjacent pixels in some spatial direction. To calculate a correlation distribution of each channel, we made a plot showing the intensity level of 1000 random pixels against their corresponding adjacent pixels in the vertical direction. In Figures 16 and 17 we show the correlation distributions for each of the color channels of image number 6, image number 15, respectively.

### 3.3. Entropy Analysis

Entropy is a scientific concept commonly used to measure a state of disorder, randomness, or uncertainty. In the case of an image encryption algorithm, it could give us information about the randomness of the pixels of the resulting image.

Thus, the entropy $H_q$ of an information source $q$ is defined as is given in Equation (14):

$$H_q = \sum_{i=0}^{R-1} p(q_i) \log_2 \frac{1}{p(q_i)},$$ (14)

where $R$ is the number of symbols $q_i$ of $q$ and $p(q_i)$ is the probability of occurrence of the symbol $q_i$. Thus, from Equation (14), the entropy for 256 equiprobable symbols corresponding to an image of 8 bits per channel or 256 intensity levels, is 8.

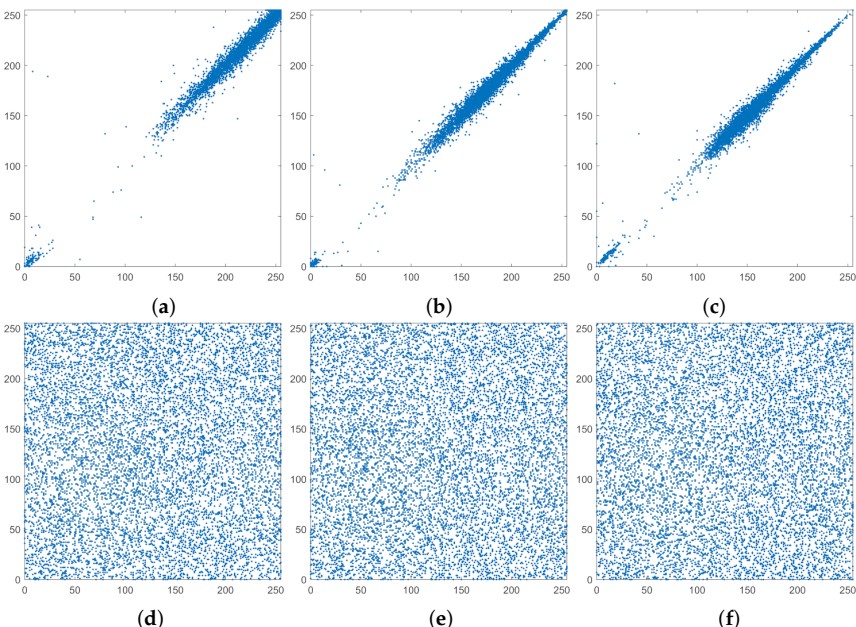

**Figure 16.** Correlation distributions of image number 6 (healthy patient). (**a**) Original image, red channel. (**b**) Original image, green channel. (**c**) Original image, blue channel. (**d**) Encrypted image, red channel. (**e**) Encrypted image, green channel. (**f**) Encrypted image, blue channel.

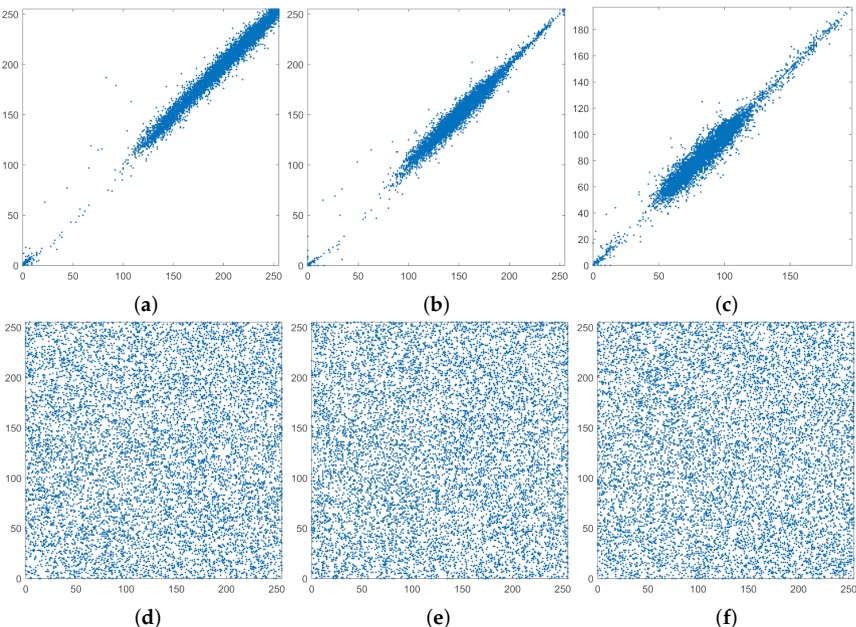

**Figure 17.** Correlation distributions of image number 15 (sick patient). (**a**) Original image, red channel. (**b**) Original image, green channel. (**c**) Original image, blue channel. (**d**) Encrypted image, red channel. (**e**) Encrypted image, green channel. (**f**) Encrypted image, blue channel.

For our algorithm applied in image number 6 (healthy patient) the average entropy for its three channels was of 7.999988 and for image number 15 (sick patient) it was 7.999989.

### 3.4. Keyspace

The keyspace is defined as the universe of every possible combination of the security keys used to encrypt an image.

For the Jigsaw transform dividing the image into $M$ subsections, there are $M!$ possible JT permutations, for example for $M = 64$ subsections we obtain $64! = 1.2689 \times 10^{89}$

possible ways to decrypt the image. Thus, the keyspace for the Jigsaw transform is given by $K_J = M!$.

Regarding the deterministic noise, which is determined given the parameters $p_1, p_2, p_3$, one for each color band, and added to row $i$, the keyspace $K_N$ is unknown but larger than $256^3$.

For the cyclic permutation, the keyspace for a $X \times Y$ image is given by $K_P = XY$.

In the case of Langton's ant, the image is divide into four regions: upper left, upper right, lower left, and lower right. Each section is then divided into four sections again. If we repeat the process $p$ times, we will obtain $4^p$ regions. Then, an ant in each region, starting in a coordinate given, walks 100 steps, and it stops; the final set of coordinates will be saved and will be our decryption key. Therefore, the keyspace for a image divided in $4^p$ sections would be given by Equation (15):

$$\prod_{i=1}^{4^p} D_i,$$ (15)

where $D_i$ is the keyspace for the section $i$. Since $D_i$ is composed of three color channels and the keyspace of each channel is determined by the final position and orientation of the ant, for a section with dimensions $m \times n$, $D_i = (4mn)^3$. For simplicity we consider the case where all sections have the same dimensions. Therefore if our complete image has dimensions $X \times Y$ and is divided into $4^p$ sections, then the dimension $m \times n$ of a section would be determined as given by Equations (16) and (17):

$$m = \left\lfloor \frac{X}{2^p} \right\rfloor \text{ and}$$ (16)

$$n = \left\lfloor \frac{Y}{2^p} \right\rfloor.$$ (17)

Then, they keyspace for Langton's ant ($K_L$) of the image is given by Equation (18):

$$K_L = \prod_{i=1}^{4^p} \left( 4 \left\lfloor \frac{M}{2^p} \right\rfloor \left\lfloor \frac{N}{2^p} \right\rfloor \right)^3 = \left( 4 \left\lfloor \frac{M}{2^p} \right\rfloor \left\lfloor \frac{N}{2^p} \right\rfloor \right)^{3*4^p}.$$ (18)

Therefore, the keyspace $K$ for a $X \times Y$ RGB image is given by Equation (19):

$$K = K_J^2 K_N^2 K_P K_L.$$ (19)

Then if our image is divided into $M$ sections of $k_1 \times k_2$ pixels for the Jigsaw transforms, and divided in $4^p$ sections for Langton's ant, the final keyspace is shown in Equation (20):

$$K > \left( \frac{XY}{k_1 k_2}! \right)^2 (256^6)(XY) \left( 4 \left\lfloor \frac{X}{2^p} \right\rfloor \left\lfloor \frac{Y}{2^p} \right\rfloor \right)^{3*4^p}.$$ (20)

For example, if we use a $4224 \times 3616$ RGB picture and divide it in sections of $16 \times 16$ for the Jigsaw transform and into $4^9$ sections for Langton's ant, then $K > 1 \times 10^{1134190.38}$. Since this number was too big to be calculated with a calculator, we instead calculated the logarithm base 10 of the keyspace, which can be obtained by using the logarithm base 10 of the variables involved and the laws of logarithms and exponents, once we get the result we raise 10 to the number obtained to get the keyspace.

### 3.5. Differential Attack

The metrics of the number of pixels change rate (NPCR) and the unified average changing intensity (UACI) are commonly used to test how strong is an encryption system

against a differential attack [12]. Given a single-band image $A(x, y)$ and a single-band image $B(x, y)$ both of size $X \times Y$, the NPCR is calculated using Equation (21):

$$\text{NPCR} = \frac{\sum_{x=1}^{X} \sum_{y=1}^{Y} D(x, y)}{X \times Y} \times 100, \tag{21}$$

where

$$D(x, y) = \begin{cases} 0 & \text{if } A(x, y) - B(x, y) = 0 \\ 1 & \text{in any other case.} \end{cases} \tag{22}$$

Meanwhile, UACI is calculated using Equation (23):

$$\text{UACI} = \frac{\sum_{x=1}^{X} \sum_{y=1}^{Y} |A(x, y) - B(x, y)|}{255(X \times Y)} \times 100. \tag{23}$$

If two similar images are encrypted and their NPCR is close to 100% and the UACI is close to 33% the metrics will confirm that a small change in the initial picture lead to a considerable change in the encrypted picture [12].

To use these metrics we take an RGB picture called $A(x, y, 3)$, we chose a pixel randomly, modify the pixel and save the result as $B(x, y, 3)$. Then we encrypt both $A(x, y, 3)$ and $B(x, y, 3)$ with the same encryption key and compare the results with NPCR and UACI, taking the average results of the three color channels.

We made a hundred for each of the 20 images shown in Figure 1 with the same parameters used in Section 3. Table 1 shows the results for the complete dataset both for healthy (H) and non-healthy (NH) patients.

**Table 1.** Results for NPCR and UACI values both for healthy (H) and non-healthy (NH) patients.

| Image | Set | NPCR (%) | | | UACI (%) | | |
|---|---|---|---|---|---|---|---|
| | | Min | Max | Mean | Min | Max | Mean |
| 1 | HP | 99.575 | 99.584 | 99.579 | 33.433 | 33.464 | 33.449 |
| 2 | HP | 99.575 | 99.584 | 99.580 | 33.429 | 33.465 | 33.448 |
| 3 | HP | 99.576 | 99.587 | 99.581 | 33.432 | 33.462 | 33.450 |
| 4 | HP | 99.574 | 99.584 | 99.580 | 33.437 | 33.465 | 33.450 |
| 5 | HP | 99.574 | 99.585 | 99.580 | 33.423 | 33.457 | 33.443 |
| 6 | HP | 99.575 | 99.585 | 99.580 | 33.431 | 33.460 | 33.444 |
| 7 | HP | 99.575 | 99.584 | 99.580 | 33.424 | 33.462 | 33.446 |
| 8 | HP | 99.574 | 99.587 | 99.580 | 33.426 | 33.467 | 33.447 |
| 9 | HP | 99.574 | 99.583 | 99.579 | 33.430 | 33.462 | 33.450 |
| 10 | HP | 99.575 | 99.586 | 99.580 | 33.426 | 33.460 | 33.443 |
| 11 | NHP | 99.573 | 99.584 | 99.579 | 33.416 | 33.462 | 33.447 |
| 12 | NHP | 99.574 | 99.584 | 99.580 | 33.424 | 33.464 | 33.446 |
| 13 | NHP | 99.575 | 99.585 | 99.579 | 33.429 | 33.468 | 33.447 |
| 14 | NHP | 99.576 | 99.585 | 99.580 | 33.421 | 33.464 | 33.446 |
| 15 | NHP | 99.574 | 99.586 | 99.580 | 33.418 | 33.462 | 33.445 |
| 16 | NHP | 99.574 | 99.586 | 99.581 | 33.421 | 33.461 | 33.445 |
| 17 | NHP | 99.575 | 99.585 | 99.579 | 33.426 | 33.467 | 33.449 |
| 18 | NHP | 99.576 | 99.589 | 99.581 | 33.427 | 33.467 | 33.448 |
| 19 | NHP | 99.575 | 99.583 | 99.579 | 33.429 | 33.464 | 33.450 |
| 20 | NHP | 99.533 | 99.582 | 99.578 | 33.416 | 33.467 | 33.447 |
| MEAN | | 99.5726 | 99.5849 | 99.5796 | 33.426 | 33.4635 | 33.4469 |
| ± STD | | 0.009355 | 0.00162 | 0.000675 | 0.0056 | 0.00289 | 0.002298 |

### 3.6. Key Sensitivity

To analyze the key sensitivity we encrypted image number 15 (sick patient) with the same parameters used in previous sections and then decrypted them with a slight change in the decryption key. Then, we compare the resulting image with the original image and measure their NPCR (taking the average NPCR of the three color channels).

When we use the wrong key for the first Jigsaw transform, we get an NPCR of 97.9205%. Using the wrong key for the first deterministic noise (increasing one of the parameters by one) we get 98.2366%. Permuting one extra column in the cyclic permutation gives 99.2293%. Using the wrong key for the second Jigsaw transform gives us 99.6065%. For the second deterministic noise we get 98.2941%. Using the wrong starting positions for the ants of the last step gives us 75.5331%. The resulting images can be seen in Figure 18.

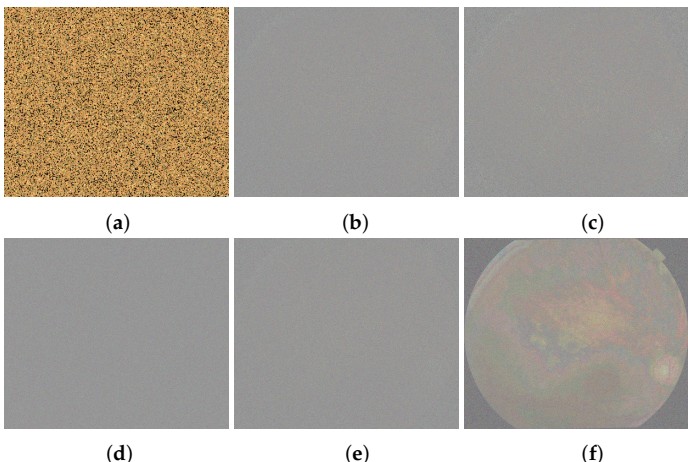

**Figure 18.** Correlation distributions of image number 15 (sick patient). (**a**) Wrong key for the first Jigsaw transform. (**b**) Wrong key for the first deterministic noise. (**c**) Wrong key for the cyclic permutation. (**d**) Wrong key for the second Jigsaw transform. (**e**) Wrong key for the second deterministic noise. (**f**) Wrong key for Langton's ant.

## 4. Discussion

In Section 3.2.1, we calculate and show the histogram from a healthy and non-healthy patient (Figures 14 and 15), where we can see that the corresponding encrypted histograms are flat in all cases, showing no similarity with the original histograms, and also, the resulting histogram of the healthy patient is indistinguishable from the one of the sick patient, making it impossible to know the condition of the person that the picture belongs to.

Regarding correlation distributions both original and encrypted images (Section 3.2.2), in Figures 16 and 17 we show that the correlation distributions for the original images shows a strong correlation between the adjacent pixels in the vertical directions (linear behavior), while the encrypted images show a weak correlation (random behavior). Similar results were obtained for other directions and for other images.

In Section 3.3, we calculated the entropy for a healthy patient obtaining an average for its three channels of 7.999988, while for a non-healthy patient, it was 7.999989. From Equation (14), for an 8-bit gray level image with 256 equiprobable symbols, the ideal entropy value would be 8. The entropy values obtained show that the proposed encryption method generates images close to a random distribution with a uniform probability density function.

From Table 1, we can see that the mean NPCR obtained was 99.5796% $\pm$ 0.000674, with values mean from 99.5726% $\pm$ 0.00935 to 99.5849% $\pm$ 0.00161. If we separate the patients into healthy and non-healthy, we obtained a mean NPCR value of 99.5796% $\pm$ 0.000489 for healthy patients and 99.5796% $\pm$ 0.000849 for non-healthy patients. Concerning the UACI value, from Table 1 we obtained a mean value of 33.4469% $\pm$ 0.00229, with minimum and maximum values varying from 33.4259% $\pm$ 0.00564 to 33.4635% $\pm$ 0.00289.

In addition, the UACI mean values were 33.4470% $\pm$ 0.00291 for healthy patients and 33.4468% $\pm$ 0.00162 for non-healthy patients.

Both for healthy and non-healthy patients, the NPCR and UACI values show that the proposed encryption approach equally hides the visual information of fundus photographs regardless of patient condition and indicating that the encrypted image is resistant to differential attacks.

In addition, the key sensitivity analysis presented in Section 3.6 shows that if we perform a small change in the key and we vary the part of the key attacked, the original image is not decrypted.

To compare our proposal, we have selected four related works based on nonlinear chaotic models. Where two proposals present general approaches, and two are focused on protecting medical images. Within the first methods, Stoyanov and Kordov [12] proposed a chaos-based image encryption scheme that uses a multiple round substitution-permutation model, which uses rotation equations and a Chebyshev map as pseudo-random bit generators, where the images used were selected from the Miscellaneous volume of the USC-SIPI image database [54]. Second, Vilardy et al. [9] presented an encryption approach using the Jigsaw transform and the iterative cosine transform over a finite field, the authors used the standard images: a woman wearing a hat, mandrill, peppers, and bridge. On the other hand, concerning the medical images cryptography methods, Moafimadani et al. [32] proposed a two-stage encryption algorithm: (i) a permutation process using the SHA-256 function and shift array circularly rule, and (ii) an adaptive diffusion. They used their RGB images acquired using the Medipix3RX chip technology, a device used today in spectroscopic imaging systems. While Javan et al. [33] defined an encryption method based on multi-mode synchronization of Chen hyper-chaotic systems applied to medical images, the authors used both standard benchmark images and their X-ray and CT images of COVID-19 patients, but they only reported the entropy values and the NPCR and UACI metrics for the CT image dataset. To show the differences of the datasets of each work, in Table 2 we show the imaging modality, the number of images, the color composition (RBG or grayscale), and the image sizes in pixels (px) of each image dataset used by the authors to be compared. We have divided the image dataset information of Table 2 into two groups. In the first group (first three rows), we include the image dataset corresponding to grayscale images, whilst in the second group (last three rows), we list the RGB image datasets. Comparing the characteristics of all image datasets, we can see that the dataset used by [12] contains the larger grayscale images, while our proposal corresponds to the highest resolution RGB images.

In Table 3, we compared our proposal with the works shown in Table 2. As evaluation metrics, we show the average entropy and the averages of the number of pixels change rate (NPCR), and the unified average changing intensity (UACI). In those cases where the authors did not report these averages, we calculate them using the published results. Although the image sizes are not the same, the metrics used do not depend on the size. Thus, entropy directly depends on the level of randomness of the pixels, and the NPCR and UACI metrics are normalized metrics by the image size. In addition, we show the keyspace domain for each method.

**Table 2.** Main modalities and differences of the image datasets used by the authors to be compared.

| Method, Year | Number, Modality | Size (px) |
| --- | --- | --- |
| [12], 2015 | 28, grayscale (natural) | $256 \times 256$ (6), $512 \times 512$ (19) $1024 \times 1024$ (3) |
| [9], 2019 | 4, grayscale (natural) | $512 \times 512$ |
| [33], 2021 | 10, grayscale (CT) | $300 \times 300$ |
| [12], 2015 | 16, RGB (natural) | $256 \times 256$ (8), $512 \times 512$ (8) |
| [32], 2019 | 4, RGB (spectroscopy) | $256 \times 256$ |
| Proposal, 2021 | 20, RGB (fundus images) | $4224 \times 3616$ |

**Table 3.** Average values results of entropy, NPCR, UACI, and the keyspace domain for the proposed method and the four comparison methods.

| Method | Entropy | NPCR (%) | UACI (%) | keyspace |
|---|---|---|---|---|
| [12] (grayscale) | 7.99769 | 99.6002 | **33.5334** | $2^{298}$ |
| [9] (grayscale) | 7.99860 | **99.6150** | 33.4900 | $(64!)^2(256^{64})$ , |
| [33] (grayscale) | 4.99529 | 99.6104 | 33.4609 | not given |
| [12] (RGB) | 7.99759 | not given | not given | $2^{298}$ |
| [32] (RGB) | 7.99957 | 99.6147 | 33.4901 | $(10^{56})(2^{128})$ |
| Proposed (RGB) | **7.99998** | 99.5796 | 33.4469 | $\mathbf{> 1 \times 10^{1,134,190.38}}$ |

From Table 2, we can see that the proposal of [12] has a dataset with more images, both grayscale and RGB. However, our proposed method has a comparable number of images, and it has the highest spatial resolution. On the other hand, from Table 3, we observe that our proposal obtains the best entropy value. Regarding the NPCR and UACI percentages, the methods of [9,12] are the best proposals, respectively, but the NPCR and UAIC values that we obtained are comparable with them. Finally, our proposal is the safest proposal, being the largest keyspace of all.

Finally, Table 4 shows a comparison of the encryption time for the works of Tables 2 and 3, where does not exist a consensus regarding the size of the image, the architecture, and the platform used to report the encryption time. Additionally, only [9] and the present work report the computer architecture, the platform, image size, and the encryption time, but for a 1024 × 1024 grayscale image in [9] and for 512 × 512 and 256 × 256 RGB images in our case. Hence it is not possible to draw conclusions concerning the fastest method.

**Table 4.** Encryption computation time comparisons.

| Method | Architecture | Platform | Size Image (px) | Time (ms) |
|---|---|---|---|---|
| [12] | 2.40 GHz Intel Core i7 | not given | 512 × 512 | 95 |
| [9] | 2.70 GHz Intel Core i7 | Matlab | 1024 × 1024 | 487 |
| [33] | not given | not given | 300 × 300 | not given |
| [12] | 2.40 GHz Intel Core i7 | not given | 512 × 512 × 3 | 290 |
| [32] | 3.4GHz Intel Core i7 | Matlab | 256 × 256 × 3 | not given |
| Proposed | 2.1 GHz AMD Ryzen 5 | Matlab | 512 × 512 × 3 | 1869 |
| Proposed | 2.1 GHz AMD Ryzen 5 | Matlab | 256 × 256 × 3 | 515.3 |

## 5. Conclusions

In this paper, we present a new image encryption and decryption algorithm. We use Langton's ant, the Jigsaw transform, and a novel deterministic noise method. Moreover, as a case of study, we applied this proposal to high-resolution retinal fundus images. The Jigsaw transform allowed hides the visual information of a picture effectively, whereas that Langton's ant process leads to a very secure and reliable approach. The proposed method is fully reversible, giving identical images (RMS equals zero) in the encryption-decryption process when the encryption key is known. In a particular way, the proposed encryption and decryption method has no problem working with big pictures.

Besides, to our knowledge, this is the first time that the Langton's ant and the Jigsaw transform have been used to encrypt fundus images efficiently and securely. On the other hand, by examining our results and comparing them with other methods, we observed that our proposal overcomes those methods concerning several and critical factors, for example, the high-resolution images handled, the entropy values and the keyspace obtained, while its performance is comparable with these methods in other metrics as uniformity of the histogram, the correlation distributions, and the NPCR and UACI values.

The analysis of the algorithm showed that it is resistant to statistical analysis techniques and that the encrypted images of sick and healthy patients are indistinguishable. A considerable advantage of our proposal is the keyspace domain, which is very large, and it far exceeds other techniques making this method extremely secure. On the other hand, according to the results of Section 3.6, Langton's ant could be the weakest part of the algorithm, but the original image is not decrypted, and only a diffuse figure is obtained. To overcome this, we can increase the number of steps the ant gives, which results in the key sensitivity could improve significantly. Further research is required for our design of the deterministic noise, to calculate its exact keyspace, analyze its weaknesses and strengths, especially because it proved to be one of the strongest parts of the algorithm.

Since this algorithm is the first time Langton's ant has been used directly on the pixels of the image to modify their value, time efficiency was not taken into consideration when writing the code. We hope this new approach can inspire more research on this method to make it more efficient or to explore similar ideas.

**Author Contributions:** Conceptualization, A.R.-A., E.M.-A., J.B., I.C.-A. and J.G.A.-C.; methodology, A.R.-A., E.M.-A., J.B., I.C.-A. and J.G.A.-C.; software, A.R.-A. and E.M.-A.; validation, A.R.-A., E.M.-A., J.B., I.C.-A. and J.G.A.-C.; formal analysis, A.R.-A., E.M.-A. and J.B.; investigation, A.R.-A., E.M.-A., J.B. and I.C.-A.; resources, I.C.-A., J.G.A.-C., M.A.H.-G. and L.M.L.-M.; data curation, I.C.-A., J.G.A.-C., M.A.H.-G. and L.M.L.-M.; writing—original draft preparation, A.R.-A., E.M.-A., J.B. and I.C.-A.; writing—review and editing, A.R.-A., E.M.-A., J.B., I.C.-A., J.G.A.-C., M.A.H.-G. and L.M.L.-M. visualization, A.R.-A., E.M.-A., M.A.H.-G. and L.M.L.-M.; supervision, E.M.-A., J.B. and I.C.-A.; project administration, E.M.-A. All authors have read and agreed to the published version of the manuscript.

**Funding:** This research has been funded by Universidad Panamericana under the Program "Fomento a la Investigación UP 2021" grant UP-CI-2021-MEX-24-ING.

**Institutional Review Board Statement:** The Ophthalmology Department of the Mexican Social Security Institute, UMAE T1-León has provided the whole image database of retinal fundus images, and the ethics approval for its use in the present research approved by a local committee under reference R-2020-1001-083.

**Data Availability Statement:** The data are not publicly available due to the privacy of patients' information.

**Acknowledgments:** Andrés Romero-Arellano, Ernesto Moya-Albor and Jorge Brieva would like to thank Facultad de Ingeniería of Universidad Panamericana for all support in this work. Ernesto Moya-Albor thanks to Program "Fomento a la Investigación UP 2021" of Universidad Panamericana for the grant UP-CI-2021-MEX-24-ING awarded. Andrés Romero-Arellano thanks to Facultad de Ingeniería of Universidad Panamericana for the incredibly generous scholarship, allowing him to continue with his studies. Besides transforming and applying class projects into formal researches. Ivan Cruz-Aceves thanks to the Mexican National Council for Science and Technology (CONACYT) for the support under project Cátedras-CONACYT No. 3150-3097.

**Conflicts of Interest:** The authors declare there is no conflict of interest.

## Abbreviations

The following abbreviations are used in this manuscript:

| | |
|---|---|
| AES | Advanced Encryption Standard |
| AT | Arnold Transform |
| CP | Cyclic Permutation |
| CT | Computer Tomography |
| DCT | Discrete Cosine Transform |
| DES | Data Encryption Standard |
| DNA | Deoxyribose Nucleic Acid |

| | |
|---|---|
| DWT | Discrete Wavelet Transform |
| HP | Healthy Patients |
| JT | Jigsaw Transform |
| LA | Langton's Ant |
| mHES | Modified High-Efficiency Scrambling |
| MSB | Most Significant Bit |
| mSPDO | Simultaneous Permutation and Diffusion Operation |
| NHP | Non-Healthy Patients |
| NPCR | Number of Pixels Change Rate |
| PWLCM | Piecewise Linear Chaotic Map |
| RMSE | Root Mean Square Error |
| ROI | Region of Interest |
| RSA | Rivest–Shamir–Adleman |
| S-box | Substitution Box |
| ToCC | Tangent over Cosine Cosine |
| UACI | Unified Average Changing Intensity |

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
