# Peer review of "Image Encryption and Decryption System through a Hybrid Approach Using the Jigsaw Transform and Langton’s Ant Applied to Retinal Fundus Images"

_axioms, doi:10.3390/axioms10030215_

Round 1

Reviewer 1 Report

The article deals with the encryption of medical images. The authors propose to use, among others, Jigsaw Transform and Langton's Ant.

The article is up-to-date and very interesting. The results of the analysis confirm that it can safely hide data in the form of medical images. The text itself is well written; it reads very well. However, a few issues require improvement or comment from the authors:

1) abstract is too long. According to the journal's template, it should contain about 200 words.
2) The related work can be expanded the more that the topic of the article is very timely, and only the query from the ScienceDirect database regarding medical image encryption in 2021 alone returns 512 records.
3) I have doubts about the practical use of the presented encryption scheme. In section 3, the authors report that the encryption/decryption process takes about 150 seconds on an average PC. Is it then effective to use such an approach in practice?
4) There is no comparison with other algorithms on the same topic.

Reviewer 2 Report

An encryption scheme for medical images has been proposed. The claims are supported with experimental results. However, I have several following comments and therefore I recommend a major revision of this submission before its publication in the journal Axiom:

-------------------------------

Comments

Section 1.

1. The literature review is not sufficient. Please review some latest schemes such as the following:

  • Saleh Ibrahim and Ayman Alharbi. Efficient image encryption scheme using henon map, dynamic s-boxes and elliptic curve cryptography. IEEE Access, 2020.
  • Naveed Ahmed Azam, Ikram Ullah, and Umar Hayat. A fast and secure public-key image encryption scheme based on mordell elliptic curves. Optics and Lasers in Engineering, 2021.
  • Dolendro Singh Laiphrakpam and Manglem Singh Khumanthem. A robust image encryption scheme based on chaotic system and elliptic curve over finite field. Multimedia Tools and Applications, 77(7):8629{8652, 2018.
  • Umar Hayat and Naveed Ahmed Azam. A novel image encryption scheme based on an elliptic curve. Signal Processing, 155:391-402, 2019.

2. There exist several efficient and secure image encryption schemes. Why those schemes cannot be used for medical images? In short please explain the difference between ordinary images and medical images. Why do we need to have some special scheme for medical images?

3. If the pixels of the plain images can have any number of bits, e.g., 24 bits images in Fig. 1, then I don’t understand why modulo 256 has been used in Eqs. (3)-(12).

4. Please add some explanation on the secret keys of the scheme in Section 2.6.

5. The reported computation time is 152.58 secs. I suggest the following:

  • Separate the running time for the encryption and decryption
  • Also discuss the running time of smaller size images since it has been claimed that the calculation time is considerably low for smaller images on page 12, 1st paragraph.

6. It is difficult to understand Table 2. Please add an explanation to address the following questions:

  • How the comparison has been done?
  • Which and of what size of images has been used?
  • How many images have been used?
  • If the image is not the same then how can we compare entropy and other results?

7. Time comparison is missing.

8. Page 1, line 3, something is wrong with the next sentences

“novel way to add deterministic noise. The Jigsaw transform allowed hides the visual information
4 of a picture effectively, whereas that Langton’s ant and the deterministic noise algorithm leads to a
5 very secure and reliable approach.”

Round 2

Reviewer 1 Report

Thanks to the authors for their answers. In my opinion, the article has gained in quality. I do not have any additional comments.

Author Response

Dear Reviewer,

Thank you very much for your kind comments and for the time spent reviewing our manuscript. All your comments and suggestions helped us a lot to improve our work.

Sincerely, 

Andrés Romero-Arellano

Dr. Ernesto Moya-Albor

Dr. Jorge Brieva

Dr. Ivan Cruz-Aceves

Dr. Juan Gabriel Avina-Cervantes

Dr. Martha Alicia Hernandez-Gonzalez

Dr. Luis Miguel Lopez-Montero

Reviewer 2 Report

The authors have greatly improved the submission and addressed all of my comments, and therefore I recommend the publication of the paper in this journal with two minor comments:

  1. The following paragraph is not correct in general, because there are several lossless image encryption schemes where one can fully recover the plain image without losing any information. So, please revise this paragraph
    On the other hand, in medical images, it is important decrypting the picture
    to get an exact copy of the original image, because often the slightest
    alteration to the image could hide potentially important medical information.
    In this sense, some encryption schemes applied to natural images do not get
    the exact original image back after decrypting it, and if we applied them to
    medical images, this could lead to a loss of critical data.”
  2. A suggestion for further: please use the same data set for comparison.
